# The Hepatic Mitochondrial Pyruvate Carrier as a Regulator of Systemic Metabolism and a Therapeutic Target for Treating Metabolic Disease

**DOI:** 10.3390/biom13020261

**Published:** 2023-01-31

**Authors:** Kyle S. McCommis, Brian N. Finck

**Affiliations:** 1Department of Biochemistry and Molecular Biology, Saint Louis University School of Medicine, Saint Louis, MO 63104, USA; 2Center for Human Nutrition, Washington University School of Medicine, Saint Louis, MO 63110, USA

**Keywords:** liver, pyruvate, mitochondria

## Abstract

Pyruvate sits at an important metabolic crossroads of intermediary metabolism. As a product of glycolysis in the cytosol, it must be transported into the mitochondrial matrix for the energy stored in this nutrient to be fully harnessed to generate ATP or to become the building block of new biomolecules. Given the requirement for mitochondrial import, it is not surprising that the mitochondrial pyruvate carrier (MPC) has emerged as a target for therapeutic intervention in a variety of diseases characterized by altered mitochondrial and intermediary metabolism. In this review, we focus on the role of the MPC and related metabolic pathways in the liver in regulating hepatic and systemic energy metabolism and summarize the current state of targeting this pathway to treat diseases of the liver. Available evidence suggests that inhibiting the MPC in hepatocytes and other cells of the liver produces a variety of beneficial effects for treating type 2 diabetes and nonalcoholic steatohepatitis. We also highlight areas where our understanding is incomplete regarding the pleiotropic effects of MPC inhibition.

## 1. Introduction

The anaerobic and aerobic metabolism of glucose liberates the stored energy of this critical nutrient to fuel myriad processes. Anaerobic glycolysis, which occurs in the cytosol, converts the six carbons in glucose to three carbon end products (pyruvate or lactate) in a stepwise fashion. Along the way, glycolysis produces a variety of intermediates with important roles in building new biomolecules but produces relatively little ATP compared to the mitochondrial oxidation of pyruvate. The synthesis of pyruvate from phosphoenolpyruvate by pyruvate kinase is an irreversible reaction and entry of pyruvate into the mitochondrial matrix is a required step for further metabolism of this biomolecule.

In the mitochondrion, pyruvate can either be oxidized by the pyruvate dehydrogenase (PDH) complex to form acetyl-CoA or is carboxylated by pyruvate carboxylase (PC) to form oxaloacetate (Figure 1). In many types of cells, the oxidation of pyruvate is an important source of usable energy by generating reducing equivalents in the TCA cycle that can be used by the electron transport chain to generate ATP [1]. However, in some cell types, including hepatocytes, pyruvate carboxylation is a significant or even primary fate of mitochondrial pyruvate. Generation of oxaloacetate is an anaplerotic reaction to maintain TCA cycle carbon integrity despite persistent carbon exit for biosynthetic reactions such as gluconeogenesis (the synthesis of new glucose), de novo lipogenesis (synthesis of new fatty acids), or the urea cycle. In either case, the import of pyruvate into the mitochondrion is required since both the PDH and PC enzymes are exclusively localized to the mitochondrial matrix.

The inner mitochondrial membrane (IMM) is an impermeable barrier for most small molecules, including solutes and most ions, which allows metabolic compartmentalization and forms the basis for the electrochemical gradient required for ATP synthesis. Many of these metabolites/ions are transported by a large family of structurally-related transporters known as the SLC25A family [2]. For several years, the mitochondrial pyruvate carrier (MPC) was deemed obligatory for pyruvate import into the mitochondrial matrix and had been studied biochemically; however, attempts to purify and identify the protein(s) that made up the MPC were unsuccessful [3]. In 2012, two groups co-published that two proteins, now known as MPC1 and MPC2, comprised the MPC in *Mammalia* and demonstrated that these proteins were both necessary and sufficient to mediate mitochondrial pyruvate import [4,5]. Interestingly, neither of the genes encoding MPC1 nor MPC2 (*Mpc1* and *Mpc2* in mice; *MPC1* and *MPC2* in humans) exhibit homology with or belong to the SLC25A family of transporters, partly explaining the prior difficulties in MPC identification. Members of the SLC16A family catalyze the import of pyruvate across the plasma membrane and there is evidence of mitochondrial localization of some SLC16A proteins to the IMM [6,7,8]. However, to our knowledge, there is little experimental evidence that SLC16A or other transporters play an important role in mitochondrial pyruvate import. One SLC25A member, SLC25A8, or uncoupling protein 2 (UCP2), may indirectly regulate mitochondrial pyruvate transport by its ability to transport carboxylic acids. While UCP2 does not transport pyruvate itself, its transport of malate, oxaloacetate, and aspartate [9] has been shown to decrease glucose oxidation and enhance lactate production in HepG2 human hepatoma cells [10]. Thus, UCP2 may regulate MPC activity to drive this glycolytic phenotype. The important discovery of the MPC proteins facilitated an abundance of subsequent studies conducted over the past decade using molecular genetic approaches to understand the role of this transporter complex in intermediary metabolism in a variety of cell types.

MPC1 and MPC2 are relatively small proteins (12 and 14 kDa, respectively) that form a heterodimer in the IMM [11]. Although some studies with recombinant proteins in vitro have suggested that MPC homodimers might have weak transport activity [12], a preponderance of evidence has suggested that endogenous MPC heterodimers are the primary form of the MPC, mediate the high-capacity transport of pyruvate, and are required for MPC complex stability and activity [11]. Indeed, genetic deletion or knockdown of one MPC protein leads to the degradation of the heterodimeric partner and knockouts of MPC1 or MPC2 are generally considered to be doubly deficient in MPC proteins and activity [4,5,13,14,15]. Germline deletion of either *Mpc* gene results in early embryonic lethality in mice [16,17,18,19]. In people, rare mutations in MPC1 and MPC2 have been identified, and these also frequently result in significant growth and developmental delay and lethality in utero or in childhood [20,21,22]. The primary phenotypes of loss-of-function MPC mutations consist of lactic acidosis, hypotonia, dysmorphia, and encephalopathy, and patients with these variants are usually severely ill [20,21,22]. Conditional loss of the MPC in the heart [15,23,24] and the central nervous system [25] in mice leads to progressive cardiomyopathy and sensitivity to seizures, respectively. In contrast, deletion of the MPC in many other types of cells is well-tolerated and may actually protect against the development of a variety of disease states. For example, as will be discussed below, mice with hepatocyte-specific deletion are protected from diabetes and progression to nonalcoholic steatohepatitis (NASH) [13,14,26,27]. Deletion in skeletal myocytes rendered mice protected from obesity [28]. Furthermore, partial MPC inhibition by pharmacological means has been proposed as a potential therapeutic approach for treating a variety of chronic disease states. In this review, we have chosen to focus our attention on the current state of our knowledge regarding the regulation of hepatic intermediary metabolism by MPC and discuss the potential therapeutic effects of targeting this carrier in cells of the liver.

## 2. Development of Specific MPC Inhibitors

Existing drugs or compounds that inhibit MPC that are currently in clinical trials fall within the thiazolidinedione (TZD) class of insulin-sensitizing drugs. Clinically approved TZDs, including pioglitazone and rosiglitazone, induce profound improvements in diabetes parameters, such as reduced glycemia and insulinemia, but their clinical use is limited by several side effects such as weight gain, fluid retention, and bone density loss. Mechanistic studies have suggested that the side effects of TZDs are due their canonical mechanism of action as ligands that activate the nuclear receptor peroxisome proliferator-activated receptor gamma (PPARγ) [29,30,31,32]. However, TZDs are also known to directly bind and inhibit the MPC [14,33,34,35]. The superior efficacy and safety of pioglitazone, which is a much weaker agonist of PPARγ compared to rosiglitazone, has caused many to hypothesize that specific MPC inhibitors that lack PPARγ binding would maintain beneficial metabolic effects with an improved safety and toxicity profile. MSDC-0160, MSDC-0602, and PXL065 are all TZD compounds in clinical development with dramatically reduced PPARγ binding/agonism in vitro compared to pioglitazone or rosiglitazone but maintain the ability to bind and inhibit the MPC [34,35,36]. These compounds display anti-diabetic and insulin-sensitizing effects in both rodent models and humans [35,36,37,38,39]. Table 1 summarizes the clinical trials for these MPC-specific agents in clinical development. However, it is difficult to confirm whether PPARγ agonism is truly spared with these TZD compounds due to the metabolism of TZDs to other active compounds in vivo.

We, and others, have begun attempts to identify or design MPC inhibitors devoid of the TZD ring. Historically, α-cyanocinnamate derivatives, such as UK-5099, are known to be potent MPC inhibitors, but these compounds have low bioavailability due to plasma protein binding and have thus been limited predominantly to use as in vitro tool compounds. Originally, it was believed that UK-5099 and other cyanocinnamates formed covalent bonds with cysteines present in the proteins making up the MPC complex [42]. However, more recently, this has been challenged by other data that mutation of all cysteines in the MPC proteins does not impair inhibitor binding, but rather these inhibitors interact by forming hydrogen bonds [43,44].

The crystal structure of the MPC has not been reported; however, using this new information and homology modeling, we developed a theoretical model of the MPC structure and identified a putative substrate-binding cavity in the MPC dimer. When three amino acid residues (Phe66 (MPC1) and Asn100 and Lys49 (MPC2)) were mutated, the binding of pyruvate and several inhibitors was attenuated [43]. This pharmacophore model for MPC inhibition allowed us to use the UK-5099 chemical scaffold to create five new MPC inhibitors with MPC inhibition potencies similar or better than UK-5099 [43]. These new inhibitors are predicted to be orally available with acceptable pharmacokinetic profiles based on their chemical properties. Two other recent studies have also modified the UK-5099 scaffold and created potent novel MPC inhibitors [44,45]. However, whether these novel compounds provide beneficial metabolic effects in vivo has yet to be tested.

In another recent study, using the bioluminescent resonance energy transfer assay for MPC binding [46], we performed a high-throughput screen of the Pharmakon 1600 library to identify existing drugs with MPC binding and potential inhibition activity. In total, this screen identified 35 compounds capable of binding MPC [47]. This screen confirmed several known MPC inhibitors such as the TZDs rosiglitazone and pioglitazone, but also identified several quinolone antibiotics as MPC binding [47]. These quinolones, the non-steroidal anti-inflammatory agent carsalam, and compounds sold as monocarboxylate transporter inhibitors 7ACC1 and 7ACC2 all share a similar pharmacophore and were all shown to inhibit mitochondrial pyruvate respiration [47]. There is broad interest in identifying novel MPC inhibitors with oral availability and good pharmacokinetic properties for treating a variety of chronic diseases.

## 3. Cells of the Liver

A variety of types of the cells can be found in the liver and work in concert to ensure that this organ, which is critical for maintaining metabolic and organismal homeostasis, functions properly. These cells are organized into liver lobules; hexagonal-shaped functional units arrayed around the central veins (Figure 2). At the edges of the lobule are the portal triads composed of the portal vein, the hepatic artery, and bile ducts. Blood from the portal vein, which comes to the liver from the small intestine, and the hepatic artery mixes in liver sinusoids and then flows into the lobule and exits through the central vein. As the oxygen- and nutrient-rich blood enters and transits the lobule, the cells housed therein perform their functions to regulate organismal homeostasis.

Hepatocytes are the parenchymal cells of the liver that perform many of the functions ascribed to the liver. These cells play crucial roles in regulating (1) systemic glucose, amino acid, fatty acid, and cholesterol metabolism; (2) detoxification of the blood; (3) dietary fat digestion and (4) the secretory functions of the liver. While 60–70% of the cells of the liver are hepatocytes, these cells make up 80% of the liver volume due their relatively large size compared to other cell types [48]. Hepatocytes exhibit heterogeneous phenotypes depending upon their location in the lobule [49]. For instance, hepatocytes in zone 3, near the portal triad, are exposed to higher concentrations of oxygen and nutrients due to their proximity to the hepatic artery and portal vein and exhibit high rates of fatty acid oxidation and gluconeogenesis (Figure 2). Hepatocytes in zone 1, adjacent to the central vein, rely more heavily on anaerobic glycolysis and have higher rates of lipid synthesis and secretion of lipoprotein particles. zone 2 hepatocytes, which are located between zones 1 and 3, exhibit an intermediary metabolic phenotype and proliferate rapidly in response to liver injury to regenerate and replace hepatocytes that are lost.

Hepatic stellate cells (HSC) are fibroblasts of the liver. In the absence of liver injury, HSC are quiescent, reside in the perisinusoidal space (Figure 2), and serve as a significant storage depot for vitamin A “retinoid” compounds in large lipid droplets that are characteristic of this cell type. However, in response to a variety of injurious stimuli, HSCs become activated, and the lipid droplets undergo lipolysis releasing their retinyl ester and triglyceride contents. Stellate cells can then migrate to the site of injury and differentiate into myofibroblasts that secrete components of the extracellular matrix that make up fibrotic lesions [50]. Although the function of HSCs is critical for the reparative processes of the liver, in some circumstances, hepatic fibrosis can become so extensive that it disrupts the normal architecture of the liver and leads to liver failure.

Hepatic macrophages play important roles as effector cells of the immune system to eliminate bacteria escaping the gut through the portal vein. Macrophages are also involved in in maintaining liver homeostasis by phagocytizing dead or dying cells to facilitate tissue repair and secreting factors that communicate with other cells of the liver in response to various stimuli. The intrahepatic macrophage population includes both a tissue resident population, known as Kupffer cells, and circulating monocytes that differentiate into macrophages after entering the liver.

The other major cell types in the liver are biliary epithelial cells (cholangiocytes) and liver sinusoidal endothelial cells. Hepatic cholangiocytes line the ducts of the biliary tree and modify the bile originally formed by hepatocytes. Interestingly, primary biliary cirrhosis is an autoimmune disease of the cholangiocytes involving the generation of anti-mitochondrial antibodies reacting to an aberrantly overexpressed E2 pyruvate dehydrogenase subunit [51]. However, overall, very little is known about pyruvate metabolism in cholangiocytes, and therefore, the implications of MPC activity in these cells will not be discussed.

It is generally believed that endothelial cells generate ATP primarily from glycolysis; however, a study in isolated liver endothelial cells suggested glycolysis was responsible for <20% of the ATP formed while glutamine and fatty acid oxidation supplied the bulk of the ATP [52]. Another recent publication confirmed that liver sinusoidal endothelial cells indeed displayed low glycolytic rates, and relatively high mitochondrial respiration when provided media containing glucose, pyruvate, and glutamine. It was concluded that these metabolic phenotypes were critical for endothelial cell scavenging and antigen presentation [53]. These studies suggest that the MPC is relatively unabundant in liver sinusoidal endothelial cells, but to our knowledge, no studies dedicated to the importance of the MPC in these cells exist.

## 4. MPC in Hepatocytes

Hepatocytes are the hepatic parenchymal cells, and as such, perform many of the metabolic functions ascribed to the liver. These cells play vital roles in regulating blood glucose levels by the uptake, storage, and release of glucose depending upon the nutritional state of the organism. Hepatocytes are also one of the few cells of the body that store glycogen for release as glucose into the blood and can synthesize glucose from other nutrients, including lactate/pyruvate, several amino acids, and glycerol. Hepatocytes can also convert these nutrients into de novo-synthesized fatty acids through the process of de novo lipogenesis. Additionally, they package and secrete fatty acids in triglyceride-rich lipoprotein particles and take up lipoprotein remnants to control the circulating concentrations of triglycerides and cholesterol. As noted in the introduction, hepatocytes are heterogeneous, and their zonal location impacts the relative rates of flux through a given pathway. However, the mitochondrial metabolism of pyruvate is central to most of these metabolic processes and recent single-cell RNA-seq analyses have suggested that the genes encoding the MPC subunits are highly expressed in hepatocytes across all zones of the lobule [54].

Although global MPC knockout results in embryonic lethality, development of conditional MPC alleles allowed mice with hepatocyte-specific knockout of *Mpc1* and *Mpc2* to be generated and characterized [13,14]. Deletion of either *Mpc1* or *Mpc2* was found to result in loss of the other MPC protein, and thus, these mice were both essentially double MPC knockouts. Consistent with this, *Mpc1* and *Mpc2* hepatocyte knockout mice exhibited remarkably similar phenotypes. Both lines of knockout mice were viable and were outwardly indistinguishable from littermate controls. Hepatic mitochondria from MPC deficient mice exhibited reduced pyruvate metabolism; however, mitochondrial membrane potential, ultrastructure, and respiration with other substrates were all normal.

### 4.1. Effects of MPC Inhibition on Hepatic Gluconeogenesis and Diabetes

Both groups characterizing the hepatocyte MPC knockout mice began by characterizing the effects of MPC deficiency on gluconeogenesis, since pyruvate and lactate are important substrates for gluconeogenesis and mitochondrial pyruvate import was thought to be critical for pyruvate entry into the gluconeogenic pathway [55]. Indeed, mitochondrial pyruvate carboxylation is a critical early step in the process of hepatic gluconeogenesis from pyruvate/lactate [56]. A variety of stimuli associated with increased rates of gluconeogenesis, including fasting, diabetes, and glucagon stimulation, led to increased hepatic expression of the genes encoding MPC proteins [13,14,57]. Loss of the MPC in hepatocytes led to a slight increase in fasting blood lactate concentrations and was shown to impair the flux of ^13^C-pyruvate into newly synthesized glucose as well as several TCA cycle intermediates [13,14], confirming that the MPC was a primary route of pyruvate entry into the mitochondrion in hepatocytes. Mice with hepatic MPC deficiency exhibited normal blood glucose concentrations in the fed state but became mildly hypoglycemic when challenged with a prolonged fast; this is consistent with an important role for hepatic pyruvate-mediated gluconeogenesis in fasted conditions.

Interestingly, neither MPC knockout mouse line exhibited severe hypoglycemia, which was likely explained by glucose production from other gluconeogenic tissues (kidney and small intestine) and metabolic compensations occurring in hepatocytes. Indeed, isotopomer tracing studies conducted with MPC-deficient hepatocytes indicated that a significant quantity of ^13^C-labeled pyruvate was still entering into the TCA cycle and incorporated into newly synthesized glucose. This could be explained, at least in part, by pyruvate-alanine cycling mediated by the alanine transaminase enzymes, ALT1 and ALT2, which interconvert pyruvate and alanine in the cytosol and mitochondrial matrix, respectively (Figure 1) [14,19]. Since alanine can be transported into the mitochondrion independently of the MPC, hepatocytes were able to partially circumvent the loss of the MPC by this pathway. Gray and colleagues [13] also showed that loss of the MPC resulted in increased contributions of glutamine to gluconeogenesis, since glutaminolysis is also an anaplerotic input into the TCA cycle that can support this process (Figure 1).

Hyperglycemia in both type 1 and type 2 diabetes is exacerbated by inappropriately high rates of hepatic glucose production, including increased synthesis of glucose from pyruvate [58]. Mice lacking the MPC in hepatocytes were significantly protected from developing hyperglycemia in the context of insulin deficiency [14], the *db/db* genetic background [14], or after prolonged feeding of a high fat diet [13,27,47]. Mechanistic studies demonstrated that this was due to diminished hepatic glucose output and pyruvate-driven gluconeogenesis. Thus, although compensatory mechanisms prevent severe fasting-induced hypoglycemia in these mice, impaired hepatic pyruvate flux is sufficient to constrain glucose production in insulin deficiency or resistance. This suggests that targeting the MPC could be an effective treatment for diabetes. In support of this, pharmacological inhibitors of the MPC that suppress gluconeogenesis in vitro and in vivo exhibited anti-diabetic effects in studies conducted in mice [35,36,37,47,59]. Moreover, two MPC inhibitors, MSDC-0160 and MSDC-0602K, have been used in clinical trials and were found to lower fasting glucose concentrations and hemoglobin A1C in people with type 2 diabetes [38,39]. Additional MPC inhibitors, including 7ACC2, zaprinast, and nalidixic acid, were also tested acutely in diet-induced obese mice, inducing mild improvements in glycemia and glucose tolerance [47]. These improvements were not via enhanced insulin sensitivity but were predominantly due to reduced hepatic glucose production. One of these compounds, 7ACC2, did not improve glucose tolerance in liver-specific Mpc2-/- mice, further suggesting that suppression of hepatic pyruvate metabolism and gluconeogenesis was likely the main mechanism of action. These results suggest that there could be a benefit to exploring the development of liver-specific MPC inhibitors by conjugating the drug to polyethylene glycol or encapsulating the drug in a lipid-based or other nanoparticle [60]. It might also be feasible to suppress hepatic MPC expression by use of hepatocyte-targeted RNAi using *N*-acetylgalactosamine-conjugated oligonucleotides. Hepatocyte-specific MPC inhibition may not improve whole-body insulin sensitivity but could improve glycemia by suppressing hepatic glucose production and may have concomitant efficacy in treating nonalcoholic fatty liver disease. Additionally, if hepatic MPC inhibition can significantly decrease hepatocyte de novo lipogenesis, this would not only improve hepatic lipid accumulation but could result in decreased lipid accumulation in other tissues, and therefore, improve insulin sensitivity. This is important given the strong epidemiologic linkage between these two chronic diseases.

### 4.2. MPC Inhibition as a Therapeutic Approach to Treat Nonalcoholic Fatty Liver Disease

Nonalcoholic fatty liver disease (NALFD) is the accumulation of lipids in the liver parenchyma in the absence of other attributable cause. NAFLD has become the most common chronic liver disease and is still without a licensed drug therapy [61]. NAFLD encompasses a range of disease severity including the formation of steatotic liver lesions, inflammation, and fibrosis known as nonalcoholic steatohepatitis (NASH). Contracting NAFLD greatly increases the risk of developing cirrhosis or hepatocellular carcinoma (HCC); it is also highly linked to the risk of type 2 diabetes and cardiovascular disease [61]. Studies conducted in rodents have suggested that MPC inhibition or genetic deletion increases fatty acid oxidation [14] and reduced pyruvate-mediated de novo lipogenesis [19], which could be effective ways to lower intrahepatic fat content to target the root cause of NAFLD progression. Mice lacking the MPC in hepatocytes exhibited reduced liver injury and other signs of progression to NASH when placed on a high fat diet or high fat diet containing added fructose and cholesterol, which exacerbates liver injury [26,27]. The protective effect included less liver inflammation and activation of HSC, suggesting that normalization of hepatocyte metabolism attenuates the release of intercellular factors that mediate communication between hepatocytes and other cells involved in the progression to NASH. This was linked to a suppression of TCA cycle flux, which was increased by the high fat diet and may play a role in NASH [58,62,63], but was surprisingly not associated with reduced hepatic lipid accumulation in MPC-deficient livers. Similarly, pharmacological inhibition of the MPC with MSDC-0602K also protected mice from the progression from NAFLD to NASH [26,37]. Finally, MSDC-0602K was tested in a phase 2b clinical trial for patients with NASH and, although it failed to achieve its primary endpoint (histologic improvements in NASH), MSDC-0602K reduced hepatic steatosis and lowered circulating levels of transaminases and other NASH biomarkers [39]. It is possible that more potent MPC inhibitors would have greater efficacy as NASH therapeutics.

### 4.3. Mitochondrial Pyruvate Metabolism in Hepatocellular Carcinoma

Hepatocellular carcinoma (HCC) is a malignancy of the liver due to uncontrolled proliferation of hepatocytes, usually occurring in people with cirrhosis or another chronic liver disease. It is well recognized that cancer cells exhibit marked alterations in energy metabolism including high rates of anaerobic glycolysis. Studies conducted in other types of cancers have suggested that suppressed expression of the MPC proteins may exacerbate the mismatch between glycolysis and mitochondrial pyruvate/lactate metabolism [64]. Similarly, two papers have suggested that MPC expression is deactivated in HCC [65,66]. In contrast, another study found that MPC was highly expressed in HCC cells and that genetic disruption of the MPC in hepatocytes protected mice from the development of HCC in two experimental models of the disease [67]. Cancer cells are highly dependent on glutamine metabolism for the generation of glutathione and components of biomass needed for cell proliferation. Evidence provided by Thompkins and colleagues [67] suggested that inhibiting mitochondrial pyruvate metabolism increased the requirement for glutamine metabolism to support TCA cycle function, and thus limited the availability of this amino acid for other purposes. Interestingly, while there are mixed reports about the effects of MPC inhibitors as anti-cancer agents, several experimental anti-tumor drugs have been shown to act as direct inhibitors of the MPC [68,69,70]. Further investigation is needed to disentangle whether MPC inhibition holds promise as a therapeutic approach to treating HCC.

### 4.4. The MPC in Response to Hepatocyte Cellular Toxicity and Stress

As noted above, mitochondrial metabolism of pyruvate is tightly linked to mitochondrial amino acid metabolism and MPC inhibition/deletion frequently results in compensatory catabolism of amino acids, particularly alanine, glutamine, and glutamate [13,14,67,71]. This can have an impact on the availability of amino acids for their other cellular fates and functions, including the synthesis of glutathione, which is a critical antioxidant. Acetaminophen (APAP) overdose is the most common cause of acute liver failure, and it has been shown that mitochondrial pyruvate metabolism is impaired following administration of high doses of APAP [72,73,74], which could affect glutathione synthesis and mitochondrial antioxidant capacity. Indeed, mice with enhanced mitochondrial pyruvate metabolism from pyruvate dehydrogenase kinase 4 deletion display improved glutathione levels and oxidation status, decreased oxidant stress, and reduced injury after APAP [75]. Recently, the effects of MPC deficiency on acetaminophen toxicity were explored and it was determined that loss of the MPC in hepatocytes did not exacerbate liver injury following APAP administration [76]. Interestingly, however, mice that were doubly deficient in both the MPC and ALT2 exhibited markedly worse liver injury compared to wild-type mice or mice with deletion of either MPC2 or ALT2 alone. Enhanced injury in the double knockout was associated with diminished glutathione synthesis, likely due to enhanced use of glutamine/glutamate in the TCA cycle. In contrast, deletion of the MPC protected hepatocytes from developing endoplasmic reticulum (ER) stress in response to tunicamycin, which is a potent inducer of this cellular process [77]. Again, this was linked to alterations in glutathione homeostasis, glutathione oxidation status, and adaptations in mitochondrial energetics.

## 5. MPC in Hepatic Stellate Cells

HSCs are fibroblasts of the liver that are normally localized in the perisinusoidal space in close contact with endothelial and epithelial cells, as well as the macrophages that reside in this area (Figure 2). Quiescent HSCs are characterized by a large intracellular lipid droplet enriched in retinoids and serve as an important site of vitamin A storage for the body. In response to liver injury, this lipid droplet undergoes lipolysis, with the retinyl-esters and triglycerides stored within either used as an energy source for the activated HSC, or secreted. HSC activation is highly influenced by the secretion of factors from other types of cells of the liver, including hepatocytes and macrophages [50]. Activated HSCs can proliferate and migrate to the site of injury and begin secreting components of the extracellular matrix including collagens and other structural proteins that make up fibrotic lesions. HSC can also regress to a quiescent state or undergo apoptosis when the stimulus causing liver injury is resolved.

In addition to the lipolytic breakdown of the lipid droplets, HSCs also exhibit other dramatic changes in metabolism when activation is triggered [78]. The reasons for the shift in metabolism are unclear but is likely due to the high energy demands of activation (proliferation, migration, production, and secretion of extracellular matrix proteins, etc.). Recent work has suggested that glucose uptake and glycolytic rates are increased by activating stimuli like treatment with transforming growth factor β [79]. Activated HSCs also display significantly increased glucose/pyruvate oxidation rates [80]; however, other studies suggest that the bulk of increased pyruvate from glycolysis is diverted to lactate despite increased mitochondrial abundance and activity in activated HSCs [78]. In addition, activated HSCs are highly reliant on glutamine metabolism and inhibition of glutaminolysis has been shown to limit activation of these cells in culture [80,81,82,83]. Work conducted in fibroblasts from other tissues suggests that glutamine is a substrate for synthesizing proline and other amino acids that are enriched in collagen [84], and for alpha-ketoglutarate bioavailability, which can regulate chromatin remodeling required for myofibroblast formation [85]. These studies show limiting flux of glutamine to this pathway can attenuate collagen production [85,86,87]. Interestingly, production of acetyl-CoA from pyruvate oxidation plays important signaling roles to activate the expression of genes characteristic of fibroblast activation, such as those encoding collagens [88]. It has been proposed that inhibition of glutamine and pyruvate metabolism in HSC could be a potential approach to limit HSC activation and ameliorate hepatic fibrosis.

Consistent with this notion, our prior work conducted with isolated HSCs has suggested that MPC inhibition directly reduces the activation of these cells in culture [26]. More recently, we have shown that genetic deletion of MPC2 in HSC cells reduces their proliferation and activation in vitro and protects mice from HSC activation in a mouse model of NASH (unpublished observation). Furthermore, MSDC-0602K treatment of mice fed a NASH-inducing diet prevented the increased expression of markers of HSC activation in vivo [26]. We also showed that MSDC-0602K could partially reverse this activation when treatment was initiated after the diet had been fed for 16 weeks. As noted above, hepatocyte-specific deletion of MPC2 also protected mice from HSC activation on a NASH diet, suggesting that MPC inhibition in hepatocytes leads to intercellular communication between hepatocytes and HSC in this model [26]. The mediators of this communication remain to be fully determined, but we did note reduced expression of proteins known to be involved in HSC activation and partially localized the effects to the release of exosomes, which are small extracellular vesicles released by a variety of cell types. Indeed, treatment of HSCs with exosomes released from hepatocyte specific MPC2 null mice led to lower HSC activation in vitro [26]. Collectively, these findings suggest that MPC inhibition reduces HSC activation both directly in HSC and by effects mediated in other cells of the liver.

## 6. MPC in Macrophages

Liver macrophages can be broadly classified into yolk sac-derived liver resident macrophages (Kupffer cells) and bone marrow monocyte-derived macrophages [89]. Kupffer cells are typically thought to be long-lived and self-replicating. In healthy liver, Kupffer cells make up the majority of liver macrophages and are highly involved in protecting the liver from gut-derived bacteria, tissue repair, and other aspects of tissue homeostasis [89]. Monocyte-derived macrophages are typically in the minority in healthy liver, but in response to liver injury or inflammation, they are recruited to and infiltrate into the liver. For example, in NASH, monocyte-derived macrophages infiltrate the liver and may make up a significant proportion of liver mononuclear phagocytes [90]. In general, it can be said that Kupffer cells are highly involved in tissue repair and maintenance, while infiltrating monocytes tend to be more pro-inflammatory and are associated with tissue inflammation and development of fibrosis. However, within these broad classifications of macrophages, it has become apparent that many sub-populations of macrophages exist and the diverse roles that each play are still incompletely elucidated.

The metabolic phenotype of these various macrophage sub-populations varies quite dramatically and plays a role in whether these cells exhibit pro-inflammatory or pro-resolving roles in the tissue. Generally, pro-inflammatory macrophages, previously categorized as M1 macrophages, rely heavily on glycolytic metabolism for ATP production [91]. In contrast, macrophages with an anti-inflammatory, pro-resolving phenotype (previously M2 macrophages), tend to rely more heavily on mitochondrial metabolism as an energy source [91]. Evidence has emerged that modulating the metabolic profile of macrophages can influence the development of the pro- or anti-inflammatory phenotype, including studies evaluating the effects of MPC inhibition. It might be predicted that MPC inhibition would promote a pro-inflammatory phenotype by forcing macrophages to rely on anaerobic metabolism consistent with an M1 profile. However, the MPC inhibitor UK-5099 reduced inflammasome activation and the release of interleukin-1β by bone marrow-derived macrophages [92] and suppressed the LPS-induced expression of several pro-inflammatory cytokines in a macrophage cell line [93]. Interestingly, UK-5099 also suppressed “alternative” activation of macrophages and the expression of genes associated with an M2 phenotype in response to interleukin 4 stimulation [94,95]. However, it should be noted that very recent work has called into question whether the effects of UK-5099 are due to inhibition the MPC, since the effects of genetic MPC deletion do not fully recapitulate those observed with UK-5099 treatment [92]. At very high concentrations, UK-5099 can bind and inhibit the plasma membrane monocarboxylate transporter (SLC16A family), which might affect the interpretation of these results [96,97]. These findings are somewhat analogous to prior work on the effects of etomoxir, an inhibitor of fatty acid oxidation with known off target effects [98], that were not phenocopied by genetic deletion of a required enzyme, carnitine palmitoyl transferase 2, in these cells [99]. Thus, the effects of targeting the MPC in macrophages remains somewhat controversial and requires further study. Furthermore, the effects of MPC inhibition in macrophages in the context of diseases of the liver, including NASH, is unknown.

## 7. Conclusions

Hepatic mitochondrial pyruvate metabolism is critical not only for oxidative ATP production but also for maintaining TCA cycle flux and biosynthetic reactions such as gluconeogenesis. While deletion or inhibition of the MPC can result in compensation from amino acid catabolism to somewhat maintain TCA cycle flux and glycemia, overall, these compensation mechanisms are not sufficient to counteract the excess fluxes of these pathways found in obesity and diabetes. Thus, inhibition of the MPC in hepatocytes is a viable therapeutic option to control hepatic glucose production and decrease hyperglycemia. Inhibition of the MPC in other hepatic cell types, such as macrophages and hepatic stellate cells, also appears to have beneficial effects, such as decreased activation of these cells, helping to lower hepatic fibrosis in NAFLD. Hopefully the future will provide more potent MPC-specific inhibitor compounds to test in the treatment of metabolic disease.

## Figures and Tables

**Figure 1 biomolecules-13-00261-f001:**
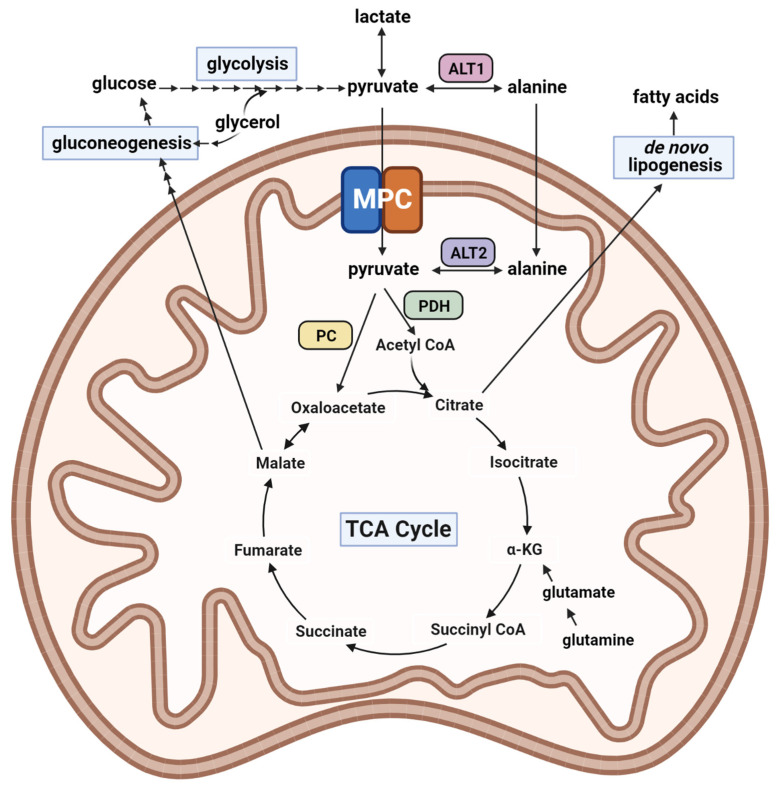
Pathways of intermediary metabolism. The schematic depicts pathways of intermediary metabolism relevant to mitochondrial pyruvate metabolism. Abbreviations: MPC, mitochondrial pyruvate carrier; ALT1, alanine transaminase 1; ALT2, alanine transaminase 2; PDH, pyruvate dehydrogenase; PC, pyruvate carboxylase; -KG, alpha-ketoglutarate. Created with BioRender.com.

**Figure 2 biomolecules-13-00261-f002:**
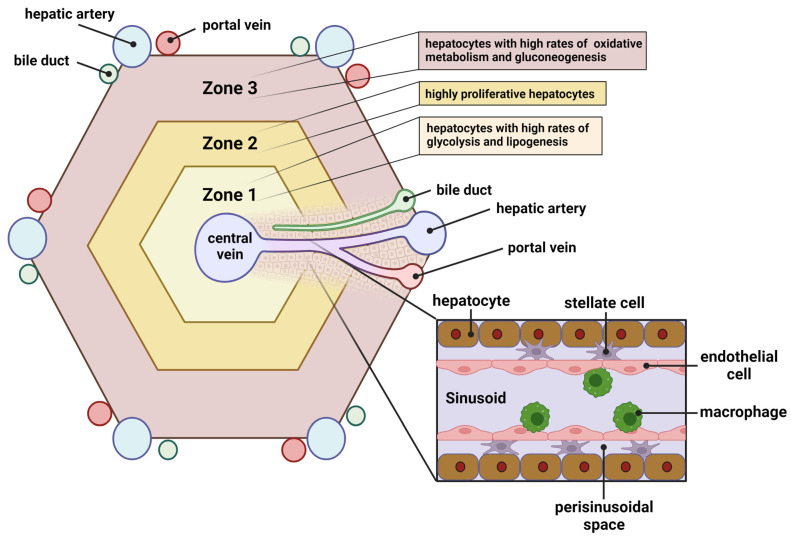
Liver lobule and hepatocyte zonation. The schematic depicts a hepatic lobule with hepatocytes residing in zones 1–3 along the length of the hepatic artery. Hepatocytes in different zones are known to display different metabolic phenotypes. Inset shows the sinusoidal and perisinusoidal space with the typical non-parenchymal resident cells. Created with BioRender.com.

**Table 1 biomolecules-13-00261-t001:** Clinical trials for MPC-specific agents in clinical development.

Compound	Stage	Dosing	Patients	Endpoints	Results	Ref.
MSDC-0160	Phase IIb	50, 100, or 150 mg vs. placebo or 45 mg Pioglitazone for 12 weeks	258 patients with diabetes	HbA_1c_ (glycemia)	100 and 150 mg doses comparable decrease to Pioglitazone	[38]
				HMW Adiponectin	100 and 150 mg doses increase, but less than Pioglitazone	
				Change in BW and waist circ.	Increased body weight, no change in waist circumference	
				Fluid retention	N.S. increased with MSDC-0160	
	Phase II	150 mg vs. placebo for 12 weeks	29 non-diabetic patients with mild Alzheimer’s	FDG-PET cerebral glucose metabolic rate	N.S. difference, but maintained over 12 weeks and decreased in placebo	[40]
				Cognitive tests	N.S. changes	
				HMW Adiponectin, hs-CRP, serum lipids	Increased HMW Adiponectin, N.S. changes in hs-CRP or lipids	
MSDC-0602K	Phase IIb	62.5, 125, or 250 mg vs. placebo for 52 weeks	392 patients with biopsy-confirmed NASH (~50% with diabetes in each treatment)	Histological improvement in NAS	30–40% improved, N.S. vs. 30% improved by placebo	[39]
				NASH resolution	20–30% improved, N.S. vs. 20% improved by placebo	
				Fibrosis improvement	24–30% improved, N.S. vs. 22% improved by placebo	
				Serum markers of glycemia/insulinemia	All doses decrease plasma glucose, insulin, HOMA-IR, and HbA_1c_	
				Serum markers of liver injury	125 and 250 mg doses decrease ALT, AST, ALP, and GGT	
				Safety	No increase in adverse events compared to placebo	
PXL065	Phase Ia	7.5, 22.5, and 30 mg or 45 mg Pioglitazone single dose	24 healthy subjects	S- and R-Pioglitazone enantiomer concentrations	22.5 or 30 mg PXL065 increases R-pioglitazone more than 45 mg Pioglitazone	[35]
				Pharmacokinetic parameters of parent drugs and active metabolites	PXL065 results in greater exposure of R-pioglitazone, but similar pharmacokinetics of active metabolites. PXL065 largely become non-deuterated (protonated).	
	Phase II	7.5, 15, or 22.5 mg vs. placebo for 36 weeks	117 patients with biopsy-confirmed NASH	Reduction in liver fat content (MRI-PDFF)	All doses improve liver fat content compared to placebo	(Prelim. Results) [41]
				Histological improvement in NAS	Not powered for histology but improvements observed compared to placebo	
				Serum markers of liver injury	ALT decreased significantly in PXL065 groups vs. placebo	
				Serum markers of glycemia/insulinemia	Glycemia, HBA_1c_, HOMA-IR, and Adiponectin improved compared to placebo	
				Safety	No dose-dependent increases in BW, low incidences of edema	

Abbreviations: ALP, alkaline phosphatase; ALT, alanine transaminase; AST, aspartate transaminase; BW, body weight; FDG-PET, fluorodeoxyglucose positron emission tomography imaging; GGT, gamma glutamyltransferase; HbA_1c_, glycated hemoglobin; HOMA-IR, homeostatic model assessment of insulin sensitivity; HMW, high molecular weight; hs-CRP, high-sensitivity C-reactive protein; MRI-PDFF, magnetic resonance imaging proton density fat fraction; NAS, Nonalcoholic fatty liver disease activity score; N.S., non-significant.

## Data Availability

Not applicable.

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
