# Peer review of "The Hepatic Mitochondrial Pyruvate Carrier as a Regulator of Systemic Metabolism and a Therapeutic Target for Treating Metabolic Disease"

_biomolecules, 2023, doi:10.3390/biom13020261_

Round 1

Reviewer 1 Report

The main topic of the review article is the mitochondrial pyruvate carrier (MPC). There are already some review articles on this topic, including one by the same authors, but in the present manuscript, the authors emphasized the role of hepatic MPC in the regulation of systemic metabolism and as a therapeutic target for the treatment of metabolic diseases. The authors studied the properties of this carrier, publishing several articles whose results were included in the present review. The review was well written and covered virtually all known data about the MPC in the liver.

Thus, the considerations that I will list below are aspects that can further improve the review:

In item 2. Cells of the liver. The authors provided an excellent description of the different cell population in the liver, the hererogeneity of the hepatic lobe with the distinctions of the periportal, perivenous and intermediate zones, and also of the participation of MPC in all cells considering their different functions and metabolism. However, I have doubts about  regarding the following sentence on  Lines 160-162" However, the mitochondrial metabolism of pyruvate is central to most of these  metabolic  processes  and  recent  single-cell  RNA-seq  analyses  have  suggested  that  the genes encoding the MPC subunits are highly expressed in hepatocytes across all zones of the lobule [27]".

I read the aforementioned article [27} and found no data on MCP unless another nomenclature was used for this carrier. Even if the expression of this protein is homogeneous in all zones, my question is whether the activity can be different in different zones due to the mechanisms that regulate its activity. I believe that the greater or lesser oxygenation as well as the type and supply of nutrients that are metabolized in pyruvate must influence its activity in different zones. At this point, a paragraph describing the regulation of MCP expression and activity is essential. If there are no studies on hepatic MCP, it would be very informative to report what is already known in other tissues.

 In item 3: MCP in hepatocytes. The entire paragraph on liver functions is excellent, but the authors did not mention hepatic metabolism of xenobiotics. This metabolism requires energy and reduction equivalents especially in the form of NADPH. MCP activity (its inhibition by potential drugs) affects the flow of glucose into the pentose pathway, with consequences for NADPH generation and glutathione and thioredoxin metabolism.

Another aspect that, in my opinion, needs to be further clarified in the article is about MCP inhibition in the treatment of diabetes. Indeed, an inhibition of MCP reduces the flow of pyruvate to gluconeogenesis and contributes to the reduction of glycemia. Hyperglycemia can be harmful to several tissues, but the other critical problem is the reduction in the rate of glucose uptake by insulin-dependent tissues. A drug that acts exclusively by inhibiting hepatic glucose production would not necessarily improve diabetes. Perhaps, in the case of diabetes associated with obesity, the possibility of lower lipid synthesis from the novo gluconeogenesis could reduce the accumulation of lipids in the liver and other tissues and by consequence reduce insulin resistance.

In the discussion started on page 272, some data on the role of MCP in controlling the mitochondrial generation of ROS would also be interesting, since  the pyruvate oxidation via TCA followed by ET  is associated with the formation of ROS and of NADPH (transhydrogenase reaction). These pathways impact the cellular redox state and the GSSHG/GHS ratio.

Author Response

In item 2. Cells of the liver. The authors provided an excellent description of the different cell population in the liver, the hererogeneity of the hepatic lobe with the distinctions of the periportal, perivenous and intermediate zones, and also of the participation of MPC in all cells considering their different functions and metabolism. However, I have doubts about regarding the following sentence on Lines 160-162" However, the mitochondrial metabolism of pyruvate is central to most of these metabolic processes and recent single-cell RNA-seq analyses have suggested that the genes encoding the MPC subunits are highly expressed in hepatocytes across all zones of the lobule [27]".

I read the aforementioned article [27} and found no data on MCP unless another nomenclature was used for this carrier. Even if the expression of this protein is homogeneous in all zones, my question is whether the activity can be different in different zones due to the mechanisms that regulate its activity. I believe that the greater or lesser oxygenation as well as the type and supply of nutrients that are metabolized in pyruvate must influence its activity in different zones. At this point, a paragraph describing the regulation of MCP expression and activity is essential. If there are no studies on hepatic MCP, it would be very informative to report what is already known in other tissues.

You are correct that different nutrient availability and oxygenation likely affects the rates of mitochondrial pyruvate oxidation. However, as discussed, pyruvate transport is also important for anaplerosis and other anabolic pathways such as de novo lipogenesis that are not directly oxygen consuming. The Mpc genes not mentioned by name in the cited RNA sequencing study. However, by inputting their gene names to the online search tool associated with the study, we found that Mpc1 and Mpc2, were expressed equivalently across all zones of hepatocytes and were not affected by circadian rhythms.

In item 3: MCP in hepatocytes. The entire paragraph on liver functions is excellent, but the authors did not mention hepatic metabolism of xenobiotics. This metabolism requires energy and reduction equivalents especially in the form of NADPH. MCP activity (its inhibition by potential drugs) affects the flow of glucose into the pentose pathway, with consequences for NADPH generation and glutathione and thioredoxin metabolism.

Thank you for this point, we have omitted discussion of xenobiotic metabolism (and NADPH/ROS) simply because there just is not data related to the MPC in these areas. -We will point out though that published and unpublished data suggests that MPC inhibition actually increases glycolytic flux as well as flux into “accessory glucose pathways” such as the pentose phosphate pathway, therefore if anything NADPH should be increased by MPC inhibition, not decreased as you suggest. -We do not mention this in the review since all available data were not derived from cells of the liver and the situation could be much different across cell types.

Another aspect that, in my opinion, needs to be further clarified in the article is about MCP inhibition in the treatment of diabetes. Indeed, an inhibition of MCP reduces the flow of pyruvate to gluconeogenesis and contributes to the reduction of glycemia. Hyperglycemia can be harmful to several tissues, but the other critical problem is the reduction in the rate of glucose uptake by insulin- dependent tissues. A drug that acts exclusively by inhibiting hepatic glucose production would not necessarily improve diabetes. Perhaps, in the case of diabetes associated with obesity, the possibility of lower lipid synthesis from the novo gluconeogenesis could reduce the accumulation of lipids in the liver and other tissues and by consequence reduce insulin resistance.

We now expand upon this in the section regarding hepatocyte-specific MPC inhibition, discussing how it will not necessarily improve whole-body insulin sensitivity. However, we were also somewhat confused by the comments that “inhibiting hepatic glucose production would not necessarily improve diabetes”. The first line anti-diabetic therapy (metformin) works by inhibiting gluconeogenesis; not by sensitizing to the effects of insulin or increasing insulin production. We were also confused by “lower lipid synthesis from the novo gluconeogenesis”. Perhaps the reviewer meant “lipogenesis” rather than “gluconeogenesis”. We now also emphasized that if hepatic MPC inhibition decreases de novo lipogenesis, both hepatic lipid accumulation and potentially whole-body insulin sensitivity could be improved.

In the discussion started on page 272, some data on the role of MCP in controlling the mitochondrial generation of ROS would also be interesting, since the pyruvate oxidation via TCA followed by ET 1 is associated with the formation of ROS and of NADPH (transhydrogenase reaction). These pathways impact the cellular redox state and the GSSHG/GHS ratio.

Similar to our response above, we do not discuss this due to complete lack of existing data regarding the hepatic MPC in ROS formation.

Reviewer 2 Report

The current article reviews the role of the MPC pyruvate carrier in the hepatocyte mitochondrion and its potential as a therapeutic target for treating hyperglycaemia.

 The review is well-written and will be of interest to those researching metabolic regulation in diabetes.  I enjoyed reading it, finding it well-informed, although  fairly narrow in scope.

 Major Comments

I think my main comment would be that I was surprised no mention of recent work on UCP2 and its impact on TCA cycle activity was made.  UCP2 may well have a role in transport of carboxylic acids across the mitochondrial membrane and influence the balance between anaerobic glycolytic activity and aerobic TCA cycle.  Since the role of MPC in managing this balance is central to the current review, I would have thought other similar discussions would be worth a reference.  The UCP2 phenomenon may have bearing on  that of MPC1/MPC2.

 The authors acknowledge that some questions have arisen regarding the specificity of MPC inhibitors and state that  UK5099 is thought to inhibit MPC but its effects are not recapitulated by MPC KO.  It would be nice to explore the modelling of this and other inhibitors and give the crystallographic evidence for binding to its target.  This might help to indicate how the lack of specificity has arisen.

 Minor Comments

The standard of writing is very high and I detected very few typos.  However, Line 298 is missing “to”.

Author Response

I think my main comment would be that I was surprised no mention of recent work on UCP2 and its impact on TCA cycle activity was made.  UCP2 may well have a role in transport of carboxylic acids across the mitochondrial membrane and influence the balance between anaerobic glycolytic activity and aerobic TCA cycle.  Since the role of MPC in managing this balance is central to the current review, I would have thought other similar discussions would be worth a reference.  The UCP2 phenomenon may have bearing on that of MPC1/MPC2.

Thank you for bringing this to our attention. We have now included a section and references about UCP2 altering pyruvate metabolism in the introduction.

The authors acknowledge that some questions have arisen regarding the specificity of MPC inhibitors and state that UK5099 is thought to inhibit MPC but its effects are not recapitulated by MPC KO.  It would be nice to explore the modelling of this and other inhibitors and give the crystallographic evidence for binding to its target.  This might help to indicate how the lack of specificity has arisen.

We have expanded this comment on page 12 to clarify. It is known that at high concentrations, UK5099 can inhibit the plasma membrane monocarboxylate transporters (MCTs; also known as SLC16A family).

The crystal structure of the MPC has not yet been experimentally resolved. Homology modeling has been used to gain understanding of how the inhibitors may interact with the MPC. We have now discussed this in the revised manuscript on page 5.

Minor Comments

The standard of writing is very high and I detected very few typos.  However, Line 298 is missing “to”.

Thank you. We have corrected.

Reviewer 3 Report

The authors have submitted an interesting and well written review about the role of MPC in the mitochondrial matrix. Although the review itself is engaging, I feel that there are some modifications that can be made to improve this review, as well as to make it of interest to a broader readership.

Major points:

1. The statement "In contrast, deletion of the MPC in many other types of cells is well-tolerated and may actually protect from the development of a variety of disease states" is very interesting, but lacks examples and citations. This is of key importance, as the aim of the review is to discuss MPC inhibition, but the entire introduction centers around how devastating MPC depletion can be. The introduction would benefit from some beneficial examples of tissue-specific MPC inhibition, with the appropriate citations.

Furthermore, this statement leads the reader to believe that the authors will discuss examples of MPC inhibition in multiple different tissues, whereas they focus the review to the liver. Therefore, ideally, the introduction would give some examples of MPC inhibition in other organs, and then state that this review will, however, focus on the liver.

2. Section 2 presupposes knowledge of the anatomy of the liver and would benefit from a figure illustrating the cell types and summarizing their roles in metabolism, perhaps with a color scale indicating high/low glycolysis vs OXPHOS, or pyruvate fate, or whichever metabolic processes the authors wish to highlight the most.

3. I think the review would benefit from being broken up a little more. For example, the first paragraph is about MPC in hepatocytes (as the title reflects), but the following three paragraphs are about MPC inhibition in mice, finally ending with three more paragraphs about MPC inhibition in three different diseases. It would be better to subdivide this section, perhaps with some slight rearranging, so that the biochemical role of MPC and the clinical relevance of MPC inhibition are in seperate, more attractively titled sections.

4. To drive home the clinical relevance of MPC inhibition, the review would benefit from a table summarizing the clinical trials and major outcomes of MPC inhibition in patients.

Right now, the review jumps from the discussion of biochemical pathways, to clinical relevance, then back to biochemistry, which I personally find a bit jarring

5. L316 - L332 is a paragraph extensively describing the effects of MPC inhibition in HSCs but always citing a single paper, which is another review. It is far more appropriate to cite the published results of the experiments here described in detail.

6. The final section becomes very technical and difficult to follow for someone who is not well versed in pharmacokinetics. It would be helpful to add a few phrases here and there to give some biological context/meaning to these statements, so that a broader readership can grasp the relevance of this section. For example, the statement "Using the UK-5099 scaffold, we recently created five new non-indole MPC inhibitors with EC50 values for MPC inhibition in the nM range" - what does this mean? Why is this good, or novel, or an improvement on the old MPC inhibitors?

Also, a curiosity regarding this section - the authors explain that MPC inhibition by TZDs have many unwanted side effects due to their interaction with PPAR. Are the unwanted toxic side effects also attributable to the inhibition of MPC in other organs where MPC inhibition is not well tolerated? What is known about this?

7. What are the efforts being made to target MPC inhibition to the liver with these drugs? It would be very interesting to expand a little bit on how researchers are developing liver-specific MPC inhibitors. 

Minor Points:

1. Is it known whether MPC is the only pyruvate importer in the mitochondrial matrix? 

2. There is a typo in L125 ("in in")

3. L141-144 is a bit of a run-on sentence, and I think there is a typo in L143 ("when provided media with" or "when provided with media with"?)

4. L210 - pharmacologic or pharmacological inhibitors?

Author Response

Major points:

  1. The statement "In contrast, deletion of the MPC in many other types of cells is well-tolerated and may actually protect from the development of a variety of disease states" is very interesting, but lacks examples and citations. This is of key importance, as the aim of the review is to discuss MPC inhibition, but the entire introduction centers around how devastating MPC depletion can be. The introduction would benefit from some beneficial examples of tissue-specific MPC inhibition, with the appropriate citations.

Furthermore, this statement leads the reader to believe that the authors will discuss examples of MPC inhibition in multiple different tissues, whereas they focus the review to the liver. Therefore, ideally, the introduction would give some examples of MPC inhibition in other organs, and then state that this review will, however, focus on the liver.

Thank you. We have added some examples of tissue specific deletion models with beneficial effects and cited the appropriate work. We have also modified the statement to indicate that this review will focus on the liver.

  1. Section 2 presupposes knowledge of the anatomy of the liver and would benefit from a figure illustrating the cell types and summarizing their roles in metabolism, perhaps with a color scale indicating high/low glycolysis vs OXPHOS, or pyruvate fate, or whichever metabolic processes the authors wish to highlight the most.

We thought this was an excellent idea and added the suggested schematic

  1. I think the review would benefit from being broken up a little more. For example, the first paragraph is about MPC in hepatocytes (as the title reflects), but the following three paragraphs are about MPC inhibition in mice, finally ending with three more paragraphs about MPC inhibition in three different diseases. It would be better to subdivide this section, perhaps with some slight rearranging, so that the biochemical role of MPC and the clinical relevance of MPC inhibition are in separate, more attractively titled sections.

We have added sub sections to this section as requested by the reviewer.

  1. To drive home the clinical relevance of MPC inhibition, the review would benefit from a table summarizing the clinical trials and major outcomes of MPC inhibition in patients.

Right now, the review jumps from the discussion of biochemical pathways, to clinical relevance, then back to biochemistry, which I personally find a bit jarring

This table is an excellent idea, and we have now added this table to summarize the clinical trials for MSDC-0160, MSDC-0602K, and PXL065. We have left out trials for pioglitazone and rosiglitazone since these compounds are much less MPC-specific. In response to the reviewer’s suggestion. We moved the section on pharmacological inhibitors to immediately following the discussion of MPC biochemistry

  1. L316 - L332 is a paragraph extensively describing the effects of MPC inhibition in HSCs but always citing a single paper, which is another review. It is far more appropriate to cite the published results of the experiments here described in detail.

The paper cited (McCommis et al. 2017 Hepatology) is the only paper we are aware of that has examined the effects of MPC inhibition on HSC activation. We also note that this reference is not a review, but a peer-reviewed research manuscript.

  1. The final section becomes very technical and difficult to follow for someone who is not well versed in pharmacokinetics. It would be helpful to add a few phrases here and there to give some biological context/meaning to these statements, so that a broader readership can grasp the relevance of this section. For example, the statement "Using the UK-5099 scaffold, we recently created five new non-indole MPC inhibitors with EC50 values for MPC inhibition in the nM range" - what does this mean? Why is this good, or novel, or an improvement on the old MPC inhibitors?

Also, a curiosity regarding this section - the authors explain that MPC inhibition by TZDs have many unwanted side effects due to their interaction with PPAR. Are the unwanted toxic side effects also attributable to the inhibition of MPC in other organs where MPC inhibition is not well tolerated? What is known about this?

We have simplified this wording so that it can hopefully be better understood by a broader audience.

This statement refers to several studies that showed that weight gain (increased adiposity), fluid retention, and bone loss in response to TZD administration were abrogated by conditional deletion of PPARgamma in adipocytes, the collecting duct of the kidney, and osteoblasts, respectively. We have now clarified this statement. To our knowledge, there is little information about MPC inhibition and toxicity.

  1. What are the efforts being made to target MPC inhibition to the liver with these drugs? It would be very interesting to expand a little bit on how researchers are developing liver-specific MPC inhibitors. 

We have now clarified that liver-targeted delivery systems could include conjugation to polyethylene glycol or encapsulation in a liposome or other targeted delivery system. We also discuss the possibility of using liver-targeted RNAi. This can be found on page 8 of the revised MS.

Minor Points:

  1. Is it known whether MPC is the only pyruvate importer in the mitochondrial matrix? 

Thank you for this question. Members of the SLC16A family catalyze the import of pyruvate across the plasma membrane and there is evidence of mitochondrial localization of some SLC16A proteins to the IMM. However, to our knowledge, there is little experimental evidence that SLC16A family proteins play an important role in mitochondrial pyruvate import. Similarly, while there were some reports of partial purification of proteins with transport activity or genetic evidence that pyruvate import was affected by deletion of IMM proteins, these turned out to be attributable to effects on other aspects of mitochondrial metabolism and not to direct import of pyruvate. To our knowledge, the MPC is the only validated mitochondrial pyruvate transporter in Mammalia.

  1. There is a typo in L125 ("in in")

Thank you. We have corrected.

  1. L141-144 is a bit of a run-on sentence, and I think there is a typo in L143 ("when provided media with" or "when provided with media with"?)

We have modified this sentence and broken it into two separate sentences.

  1. L210 - pharmacologic or pharmacological inhibitors?

Pharmacological.